# The Role of the N-Terminal of the Prohormone Brain Natriuretic Peptide in Predicting Postoperative Multiple Organ Dysfunction Syndrome

**DOI:** 10.3390/jcm11237217

**Published:** 2022-12-05

**Authors:** Piotr Duchnowski

**Affiliations:** Cardinal Wyszynski National Institute of Cardiology, 04-628 Warsaw, Poland; pduchnowski@ikard.pl

**Keywords:** NT-proBNP, multiple organ dysfunction syndrome (MODS), valve surgery

## Abstract

Background: Multiple organ dysfunction syndrome (MODS) is the progressive and potentially reversible dysfunction of at least two organ systems in the course of an acute and life-threatening disorder of systemic homeostasis. MODS is a serious post-cardiac-surgery complication in valvular heart disease that is associated with a high risk of death. This study assessed the predictive ability of selected preoperative and perioperative parameters for the occurrence of MODS in the early postoperative period in a group of patients with severe valvular heart disease. Methods: Subsequent patients with significant symptomatic valvular heart disease who underwent cardiac surgery were recruited in the study. The main end-point was postoperative MODS, defined as a dysfunction of at least two organs—perioperative stroke, heart failure requiring mechanical circulatory support, respiratory failure requiring mechanical ventilation, and postoperative acute kidney injury requiring renal replacement therapy. A logistic regression was used to assess relationships between variables. Results: There were 602 patients recruited for this study. The main end-point occurred in 40 patients. Preoperative NT-proBNP (OR 1.026; 95% CI 1.012–1.041; *p* = 0.001) and hemoglobin (OR 0.653; 95% CI 0.503–0.847; *p* = 0.003) are independent predictors of the primary end-point in a multivariate regression analysis. The cut-off point for the NT-proBNP value for postoperative MODS was calculated at 1300 pg/mL. Conclusions: A high preoperative level of NTpro-BNP may be associated with the onset of MODS in the early postoperative period. The results of the study may also suggest that earlier cardiac surgery for significant valvular heart disease may be associated with an improved prognosis in this group of patients.

## 1. Introduction

Multiple organ dysfunction syndrome (MODS) is a very serious complication that may occur in the early postoperative period in patients undergoing cardiac surgery due to valvular heart disease. It is the leading cause of death in critically ill patients [1,2,3]. MODS is a progressive and potentially reversible dysfunction of at least two organ systems in the course of the acute and life-threatening disorder of systemic homeostasis [4,5]. The mechanical support of organs, such as hemodiafiltration in the case of renal failure, mechanical ventilation in the case of respiratory failure, or mechanical circulatory support with the use of intra-aortic balloon pump (IABP) and extracorporeal membrane oxygenation (ECMO), can correct the physiological state of a patient in the multi-organ failure syndrome, improving oxygen supply to peripheral tissues and tissue metabolism, while providing the necessary time for the regeneration of individual organs, such as a central nervous system, kidneys, lungs, and heart [6,7,8,9,10]. Knowledge of the predictors of MODS occurrence in the postoperative period is essential because it can inform the preoperative selection of patients at increased risk of this complication and special care for selected patients during the perioperative period. Available research lacks information on the prediction of postoperative MODS in patients undergoing valvular heart surgery [3,11,12,13,14]. The N-terminal fragment of B-type natriuretic propeptide (NT-proBNP) is the inactive part of the prohormone proBNP. It is formed in the cardiomyocytes, mainly of the left ventricle, from which it enters the blood and plays a special role in maintaining the homeostasis of the cardiovascular system. NT-proBNP is made up of 76 amino acids and, during secretion by the left ventricular cardiomyocytes, it accompanies the active molecule of BNP, which comprises 32 amino acids. The process of the secretion of NT-proBNP and BNP peptides occurs due to the response of myocardial cells to an increase in their preload and/or afterload. Increasing the secretion of BNP and NT-proBNP causes the activation of compensatory mechanisms that occur even before the onset of symptoms of heart failure. Thus far, available research widely describes the predictive potential of NT-proBNP in numerous diseases of the cardiovascular system, such as heart failure, myocardial infarction, chronic coronary syndrome, and postcardiotomy shock [3,15,16,17,18]. However, in the more generally available literature, there are no studies on the ability of NT-proBNP to predict MODS in the early postoperative period. Therefore, the primary objective of the present study was to assess the usefulness of the preoperative level of NT-proBNP for predicting postoperative MODS in the early follow-up period after cardiac surgery in patients with severe valvular heart disease.

## 2. Methods

This prospective study was conducted at the National Institute of Cardiology in Warsaw, Poland, on a group of consecutive patients who underwent cardiac surgery for symptomatic severe valvular heart disease. The following exclusion criteria were considered in this study: lack of consent for inclusion in the study, significant atherosclerotic changes in the carotid arteries, aged under 18 years, porcelain aorta, coexistence of autoimmune diseases, inflammatory bowel diseases, and neoplastic diseases in the active stage. Each patient had a blood sample taken for biomarkers the day before surgery. A Sysmex K-4500 electronic counter was used to assess the complete blood count. NT-proBNP levels were assessed using the electrochemiluminescent immunoassays of Elecsys 2010 (Roche, Mannheim, Germany). In each case, a median sternotomy was performed under normothermia and general anesthesia, and cold-blooded cardioplegia was applied. In the present study, the main end-point was multiple organ dysfunction syndrome, which occurred in the early postoperative period. The definition of MODS includes the dysfunction of at least two organs—the central nervous system (perioperative stroke confirmed in a neurological examination and computed tomography), cardiovascular system (postoperative cardiogenic shock requiring mechanical circulatory support in the form of ECMO or IABP), respiratory failure (mechanical ventilation for more than 24 h after cardiac surgery or the need for reintubation in a patient with complete respiratory failure), or acute kidney injury (requiring renal replacement therapy—hemodiafiltration). The follow-up period lasted until hospital discharge or death. Consent for this study was obtained from the bioethics committee operating at the National Institute of Cardiology in Warsaw, Poland.

### Statistical Analysis

The IBM SPSS software, version 2.0 (SPSS Inc., Chicago, IL, USA), was used for statistical analyses. The data presented are described as the mean ± SD and frequency (%). The Mann–Whitney U test, Pearson χ^2^ test, or Student’s *t*-test were used to compare particular groups of variables. The Shapiro–Wilk normality test was used to study the distribution of the sample. The relationship between variables was assessed using logistic regression analysis. Pre- and intraoperative variables (listed in Table 1) were examined for an association with the main end-point using univariate analysis. Statistically significant variables (*p* < 0.05) obtained in the univariate logistic regression analysis were then subjected to multivariate logistic regression analysis. Using the Youden index, the cut-off point for NT-proBNP values was determined, which met the criterion of maximum sensitivity and specificity for postoperative MODS. The predictive value of NT-proBNP for the occurrence of the main end-point was estimated using the area under the receiver operating characteristic (ROC) curve. In order to investigate possible relationships between the variables, the Spearman rank correlation test was performed.

## 3. Results

A total of 602 patients were recruited in this study and underwent cardiac surgical treatment for valvular heart disease (mean age 64 ± 12 years; 58% of the study group were men). Table 1 presents the perioperative characteristics of the patients. The main end-point (postoperative MODS) occurred in 40 patients: 34 patients required hemodiafiltration, 37 patients had prolonged mechanical ventilation, and 12 patients had mechanical circulatory support (ECMO: 10 patients; IABP: 2 patients). Twelve patients had a perioperative stroke. Among all patients with postoperative MODS, 39 patients required catecholamines for more than 48 h due to persistent postoperative hemodynamic instability. The statistically significant predictors of the main end-point at univariate and multivariate analysis are presented in Table 2. In the multivariate analysis, only NT-proBNP (OR 1.026; 95% CI 1.012–1.041; *p* = 0.001) and hemoglobin levels (OR 0.653; 95% CI 0.503–0.847; *p* = 0.003) remained independent predictors of the main end-point. The cut-off point for the NT-proBNP value for postoperative MODS was calculated at 1300 pg/mL. Figure 1 shows the area under the receiver operator characteristic curve of the NTproBNP parameter for the occurrence of the main end-point, which was 0.762 (95% CI 0.726–0.800) (sensitivity: 78%; specificity: 65%). The conducted analysis showed a significant correlation between the NT-proBNP concentration and NYHA classes (r = 0.42, *p* < 0.001), LVEF (r = −0.38; *p* < 0.001), high-sensitivity Troponin T concentration (r = 0.38; *p* < 0.001), and GFR (r—0.35; *p* <0.001). Of the 40 patients who had postoperative MODS, 29 cases were fatal. The mean concentration of NT-proBNP in the group of patients with postoperative MODS who died was 6406 pg/mL (±1209) and was significantly higher compared to patients with postoperative MODS who survived this complication: 4058 pg/mL (±1208) (*p* < 0.05).

## 4. Discussion

The main pathophysiological basis of postoperative MODS is the hypoxia of peripheral tissues, which in turn leads to cell damage. Hypoxia can also be caused by three underlying causes that may coexist: reduced cardiac output, reduced hemoglobin levels, or impaired oxygen uptake by target cells [19]. In patients with MODS developing in the postoperative period, the early use of mechanical devices in the form of IABP or ECMO, hemodiafiltration, or mechanical ventilation improves the perfusion of peripheral tissues and their oxygenation, allows to compensate for volemic and metabolic disorders, as well as respiratory disorders [6,9]. Using knowledge of MODS predictors is extremely important because it enables the preoperative identification of patients at risk of postoperative multiple organ dysfunction syndrome, the special supervision of the patient during surgery and the early postoperative period, and the early intensification of treatment. This includes the use of advanced techniques for the mechanical support of individual organs, thus giving individual organs the chance to simultaneously regenerate increasing the patient’s chances of survival. In the presented study, it was shown that the preoperative NT-proBNP concentration was an independent predictor of postoperative MODS, although the preoperative hemoglobin level also reached a significant predictive value in the multivariate analysis.

Cardiac output (CO), i.e., the amount of blood pumped by the left ventricle in 1 min, next to blood count parameters, is one of the main determinants of the ability of the body to deliver oxygen to the cells. Heart failure, which is manifested, inter alia, by a decreased cardiac output is common among patients admitted to an intensive cardiac care unit after heart valve surgery [20]. In the present study, 39 out of 40 patients with postoperative MODS required catecholamines for more than 48 h due to persistent hemodynamic instability. NT-proBNP is an important peptide used in the diagnosis and assessment of the severity and prognosis of heart failure [3,21,22,23]. The left ventricular muscle of a patient with severe symptomatic valvular disease is overloaded by 1high blood pressure and/or increased blood volume, forcing the increased secretion of NT-proBNP by cardiomyocytes. The persistent long-term overload of the left ventricular muscle promotes a progressive degenerative process, including gradual necrosis as well as a fibrosis of cardiomyocytes [3,24,25]. A very high concentration of NT-proBNP in the blood of patients with severe valvular heart disease may indicate a severe overload or even decompensation of the left ventricular muscle, which is confirmed by the statistically significant correlation between NT-proBNP concentration and left ventricular ejection fraction, Troponin T concentration or functional class according to the NYHA scale. Apart from this, the significant correlation shown between the levels of NT-proBNP and GFR may indicate that increasing heart failure promotes hypoxia and damage to peripheral tissues, leading to organ failure even before the qualification of cardiac surgery. It seems that the trend towards statistical significance for the duration of extracorporeal circulation for the main end-point, as shown in the present study, is not without significance. The above may indicate that physiological reserves of individual organs in patients with advanced heart failure are limited and that those vital internal organs, such as the central nervous system, heart, kidneys, or lungs, are particularly sensitive to the unfavorable conditions of cardioplegia and extracorporeal circulation.

The results presented above indicate that the occurrence of hypoxia and related tissue damage in the perioperative period is also influenced by the baseline hemoglobin concentration, which also decreases due to the loss of perioperative blood and the use of extracorporeal circulation [11,26,27,28]. It seems, therefore, that the coexistence of postoperative hemodynamic instability (as mentioned above) in this study was present in 39 out of 40 patients who had a main end-point in a further follow-up, and decreased hemoglobin values may be decisive for the development of postoperative MODS. Therefore, taking into account the above-described aspects and the results obtained in this study, it seems that the earlier qualification of severe symptomatic valvular heart disease for cardiac surgery with less heart damage and, consequently, the lower concentration of NT-proBNP may be associated with an improved prognosis and a lower incidence of MODS in the perioperative period [3].

## 5. Conclusions

The preoperative concentration of NT-proBNP in the blood may be a useful parameter for assessing the risk of developing multiple organ dysfunction syndrome in the early postoperative period in patients undergoing cardiac surgery due to severe valvular heart disease. These are the results of a single-center study with a moderate number of patients recruited for the study. In future studies, increasing the number of research centers and participants may confirm our results. In conclusion, the assessment of the NT-proBNP concentration may be helpful for the care of patients with severe symptomatic valvular disease, both during observation and qualification for cardiac surgery and in the early perioperative period.

## Figures and Tables

**Figure 1 jcm-11-07217-f001:**
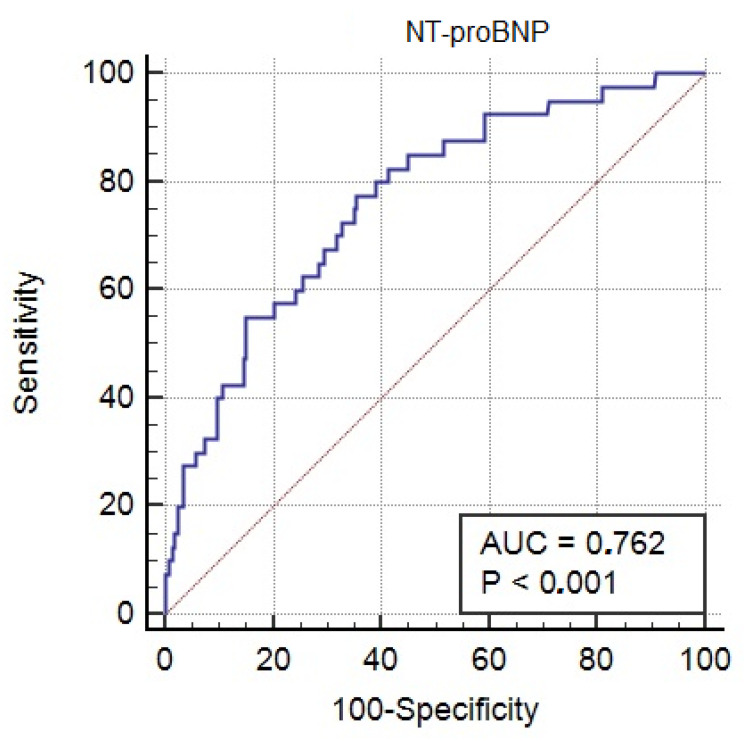
Area under the receiver operating curve for the main end-point (multiple organ dysfunction syndrome) for NtproBNP.

**Table 1 jcm-11-07217-t001:** Characteristics of the entire study group.

Patient Characteristics (n = 602)	Values
Age, years *	64 ± 11
Atrial fibrillation, n (%)	263 (43%)
Body mass index, kg/m^2^ *	27 ± 7
Diabetes mellitus, n (%)	112 (18%)
EuroSCORE II, (%) *	3.3 ± 3.0
Male: men, n (%)	347 (58%)
NYHA, (classes) *	2.6 ± 0.5
GFR, mL/min/1.73 m^2^ *	67 ± 17
Hemoglobin, g/dL *	13.7 ± 1.5
Hs-TnT, ng/L *	35 ± 27
NT-proBNP, pg/mL *	2012 ± 1504
LV ejection fraction, (%) *	58 ± 11
TAPSE, mm *	21 ± 8
RVSP, mmHg *	45 ± 18
Medications	
Antihypertensive drugs (beta blockers, ACEi, ARBs, calcium blockers, diuretics) (%), n (%)	450 (74%)
Oral antidiabetic drugs, n (%)	110 (18%)
Insulin, n (%)	22 (3%)
Statins, n (%)	222 (37%)
Anticoagulants, n (%)	263 (43%)
Intra- and postoperative characteristics of patients	
Aortic valve replacement, n (%)	302 (50%)
Aortic valve plasty, n (%)	11 (2%)
Mitral valve replacement, n (%)	108 (18%)
Aortic valve replacement + mitral valve replacement, n (%)	52 (9%)
Mitral valve plasty, n (%)	113 (18%)
Tricuspid valve plasty and replacement, n (%)	16 (3%)
Additional procedure *Coronary artery bypass* graft, n (%)	87 (14%)
Aortic cross-clamp time, min *	102 ± 31
Cardiopulmonary bypass time, min *	135 ± 56

Variables are presented as the mean * or a measure of the variability of the internal standard deviation. Abbreviations: ACEi = angiotensin-converting-enzyme inhibitors, ARBs = angiotensin receptor blockers, Hs-TnT = high-sensitivity Troponin T, NT-proBNP = n-terminal of the prohormone brain natriuretic peptide, GFR = glomerular filtration rate, LV = left ventricle, NYHA = New York Heart Association, RVSP = right ventricular systolic pressure, TAPSE = tricuspid annular plane systolic excursion.

**Table 2 jcm-11-07217-t002:** Factors associated with postoperative multiple organ dysfunction syndrome—univariable and multivariable analyses.

		Univariable			Multivariable	
Variable	Odds Ratio	95% Cl	*p*-Value	Odds Ratio	95% Cl	*p*-Value
Hemoglobin, g/dL	0.525	0.422–0.654	<0.001	0.653	0.503–0.847	0.003
NT-proBNP, pg/mL	1.022	1.011–1.033	<0.001	1.026	1.012–1.041	0.001
Age, years	1.069	1.030–1.109	0.004			
GFR, mL/min/1.73 m^2^	0.965	0.947–0.984	0.002			
LV ejection fraction, (%)	0.969	0.946–0.992	0.009			
RVSP, mmHg	1.022	1.005–1.039	0.01			
Cardiopulmonary bypass time, min *	1.040	0.972–1.121	0.08			

Abbreviations: NT-proBNP = n-terminal of the prohormone brain natriuretic peptide; GFR = glomerular filtration rate, LV = left ventricle, * the variable that reached the closest value of statistical significance (*p* < 0.05).

## Data Availability

Research data available from the author of the publication.

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
