# Peer review of "The Role of the N-Terminal of the Prohormone Brain Natriuretic Peptide in Predicting Postoperative Multiple Organ Dysfunction Syndrome"

_jcm, 2022, doi:10.3390/jcm11237217_

Round 1
Reviewer 1 Report
Dear authors,
Hi,
I would like to suggest the following changes to improve the manuscript.
1. Please note the cutoff value for NT-proBNP in the ROC curve. Also, how does that cutoff value compare to other values in the literature?
2.Please add the oral medications in Table 1. Also, how many years have you followed surgical cases over the years?
3.Please also create a multivariate analysis table in Table 2.
Best Regards
Author Response
Answers - Reviewer 1.
Thank you very much for your interesting review.
To the best of our knowledge, this is one of the first papers describing the predictive ability of the preoperative NTproBNP level in predicting MODS in the early postoperative period. Receiver operating characteristic analysis determined the cut-off value of NT-proBNP for the prediction of the occurrence of the postoperative MODS above 1300 pg/mL (area under curve = 0.762, p < 0.05) (Figure 1).
As suggested by the reviewer, the table was supplemented with medications taken.
I have been observing patients undergoing cardiac surgery for about 15 years.
As suggested, Table 2 was supplemented with the results of the multivariate logistic regression analysis.
Thank you very much.
Reviewer 2 Report
The authors conducted a prospective cohort study in 602 patients with significant valvular heart disease and receiving elective valve surgery. Results showed that increased NT-proBNP level was associated with higher risk in postoperative MODS. However, I have a few comments as below:
1. I suggest that the authors should modify their wordings and correct typos, such as “… in a group of patients undergoing heart valve” (Page 2, Line 64)
2. I notice that the author found at least 6 predictors, including hemoglobin, NT-proBNP, age, GFR, LV ejection fraction, RVSP. Among them, NT-proBNP was associated with GFR. Also, we know that NT-proBNP is associated with LV ejection fraction. Could the author please clarify why NT-proBNP is more important than other predictors?
3. In the 40 patients with postoperative MODS, 34 had renal replacement therapy. Lower GFR instead of NT-proBNP might be a better predictor to explain the observation. Please clarify.
Author Response
Answers - Reviewer 2.
Thank you very much for your review.
- As suggested by the reviewer, selected expressions were corrected.
- Given the important problem of multiple organ failure syndrome - a complication with a high risk of death - which may develop in the early postoperative period, the aim of the study was to identify predictors of this complication. Wide range of parameters analyzed. And as Table 2 shows, the univariate analysis revealed several predictors, including NtproBNP and GFR, but in the multivariate analysis, only NtproBNP and hemoglobin level were independent predictors of postoperative MODS.
Among all patients with postoperative MODS, 39 patients required supply of catecholamines for more than 48 hours. NT-proBNP is a prohormone secreted into the blood by cardiomyocytes (mainly the left ventricle). Due to the fact that the active form of BNP is actively involved in maintaining cardiovascular homeostasis, NT-proBNP is currently widely used in the diagnosis and progression of heart failure. In the severe valvular heart defects, there is a pressure and/or volume overload of the left ventricle muscle, which leads to an increase in NT-proBNP secretion by cardiomyocytes. Prolonged left ventricular wall overload is the cause of the progressive myocardial degenerative process involving gradual cardiomyocyte necrosis and fibrosis. Very high NT-proBNP values ​​present in the blood serum of patients with hemodynamically significant valvular heart disease may indicate the decompensation of an overloaded left ventricular muscle. which is indicated by a significant correlation between the level of NTproBNP and left ventricular ejection fraction, functional class according to the NYHA scale or also the preoperative level of Troponin determined by the high sensitivity method. On the other hand, the significant correlation shown between the levels of NTproBNP and GFR may indicate that increasing heart failure promotes hypoxia and damage to peripheral tissues, leading to organ failure even before qualification for cardiac surgery. All this indicates that perioperative tissue hypoxia and, consequently, the development of multi-organ failure may, to a large extent, result from the efficiency of the heart in the periprocedural period, of which the level of NtproBNP may be a very good determinant.
As mentioned above, the performed analyzes showed a significant correlation between the preoperative level of NtproBNP and GFR. This may indicate that already in the preoperative period, heart failure adversely affects the function of the kidneys, causing their hypoxia and impaired function. It may also indicate declining reserves for further hypoxia/injury that may occur during non-physiological conditions during extracorporeal circulation, when the hematocrit (as a result of blood dilution) decreases and blood pressure decreases. And as Table 2 shows, in the univariate logistic regression analysis, GFR appears next to NtproBNP, but in the multivariate logistic analysis, among the selected independent factors, MODS predictors in the multivariate analysis were only NtproBNP and hemoglobin level.
Thank you very much.
Round 2
Reviewer 2 Report
Thank you for the authors' reply. I notice that the authors had responsed the comments and refined the manuscript. But it seems that the authors simply copied several sentences from the manuscript, which might not fully address the comments. Also, I have a few comments as below:
1. I would suggest the authors to very carefully review the manuscript and refine the wording. Some typos still exist. E.g., "postoperavite" (Table 1), "NTproBNP" (Line 232)...
2. I would suggest to report important index when performing ROC analysis. E.g., sensitivity, specificity, accuracy, NPV, PPV, ... It would be clinically useful. Also, I would suggest the authors to show how the cut-off value was generated.
3. I think theoretically in multivarible analyses, one variable may need to be independent from others. Clearly, EF, NT-proBNP, GFR are correlated. Statistical review might be needed.
4. In "statistical analyses", the authors stated that "predictive value of NT-proBNP was assessed by a comparison of the areas under the receiver operative characteristics of the respective curve". I do not understand which variable was compared with NT-proBNP. No results were present.
Author Response
Thank you very much for your review.
Reference 1. Text re-examined and typos corrected again.
Reference 2. The cut-off point for the NT-proBNP value, which fulfilled the criterion of maximum sensitivity and specificity for postoperative MODS, was determined based on the Youden index. The cut-off point for the NT-proBNP value for postoperative MODS was calculated at 1300 pg/mL. The area under receiver operator characteristic curve for primary end-point for NT-proBNP is 0.762 (95% CI 0.726–0.800) (sensitivity: 78%; specificity: 65%).
Reference 3. The aim of the study was to search for predictors of the occurrence of postoperative multiple organ failure syndrome among many listed in the methodology of the article. The factors identified by the use of logistic regression analysis are presented in Table 1. Of course, the human body is a whole and therefore certain parameters correlate with each other to a greater or lesser extent - among others, the patient's age as well as the parameters of the circulatory system correlate with parameters of kidney function or parameters hematology, etc., however, as described in the study methodology, appropriate statistical methods were used. Thus, the performed univariate statistical analysis identified variables that were then included in the multivariate analysis. In the multivariate analysis, only two parameters among the subjects turned out to be independent for the occurrence of postoperative multiple organ failure syndrome: NT-proBNP and hemoglobin level. These are the results of the study and on this basis the presented article was created. The results of the presented study indicate that the occurrence of hypoxia and related tissue damage in the perioperative period is also influenced by the initial hemoglobin level, which additionally decreases due to the loss of perioperative blood and the use of extracorporeal circulation. It seems, therefore, that the coexistence of postoperative haemodynamic instability (as mentioned above in the presented study was present in 39 out of 40 patients who had the primary endpoint in further follow-up) with decreased hemoglobin values may be decisive for the development of postoperative MODS.
Reference 4. The predictive value of NT-proBNP for the occurrence of the primary endpoint was estimated using the area under the receiver operating characteristic (ROC) curve.
I am asking for the possibility of further processing of the article.
Thank you.
